# High-Power Ultrasound in Enology: Is the Outcome of This Technique Dependent on Grape Variety?

**DOI:** 10.3390/foods12112236

**Published:** 2023-06-01

**Authors:** Paula Pérez-Porras, Encarna Gómez Plaza, Leticia Martínez-Lapuente, Belén Ayestarán, Zenaida Guadalupe, Ricardo Jurado, Ana Belén Bautista-Ortín

**Affiliations:** 1Department of Food Science and Technology, Faculty of Veterinary Science, University of Murcia, Campus de Espinardo, 30100 Murcia, Spain; paula.perez2@um.es (P.P.-P.); anabel@um.es (A.B.B.-O.); 2Institute of Vine and Wine Sciences, ICVV (University of La Rioja, Government of La Rioja and CSIC), Finca La Grajera, 26007 Logroño, Spain; leticia.martinez@unirioja.es (L.M.-L.); belen.ayestaran@unirioja.es (B.A.); zenaida.guadalupe@unirioja.es (Z.G.); 3Agrovin S.A., Av. de los Vinos, s/n, Alcázar de San Juan, 13600 Ciudad Real, Spain; rjurado@agrovin.com

**Keywords:** ultrasound, grape, wine, anthocyanins, tannins, polysaccharides

## Abstract

The disruptive effect exerted by high-power ultrasound (US) on grape cell walls is established as the reason behind the chromatic, aromatic and mouthfeel improvement that this treatment causes in red wines. Given the biochemical differences that exist between the cell walls of different grape varieties, this paper investigates whether the effect of the application of US in a winery may vary according to the grape variety treated. Wines were elaborated with Monastrell, Syrah and Cabernet Sauvignon grapes, applying a sonication treatment to the crushed grapes using industrial-scale equipment. The results showed a clear varietal effect. The wines made with sonicated Syrah and Cabernet Sauvignon grapes showed an important increase in the values of color intensity and concentration of phenolic compounds, and these increases were higher than those observed when sonication was applied to Monastrell crushed grapes, whereas Monastrell wines presented the highest concentration in different families of polysaccharides. These findings correlate with the differences in the composition and structure of their cell walls since those of Monastrell grapes presented biochemical characteristics associated with a greater rigidity and firmness of the structures.

## 1. Introduction

The interest of the enological industry in improving the extraction of compounds of interest during the maceration of red wines has led to the use of classical technologies, such as the use of enological enzymes, and to the incorporation of innovative technologies such as pulse electric fields and high-power ultrasound (US) in the production processes. In recent years, the use of high-power ultrasound has resulted an effective tool for obtaining wines with better chromatic characteristics and higher content of phenolic, polysaccharide and aromatic compounds [1,2,3]. Although its use in wineries was approved in 2019 [4], there are still few studies that have been carried out with industrial equipment instead of small laboratory equipment. The use of large equipment may offer results that can be extrapolated to any winery. 

The improvement of the extraction of phenolic, polysaccharide and aromatic compounds during the maceration of grapes is based on the breakage of the cell walls of the skin cells, which are natural barriers against the diffusion of the compounds of interest located inside these cells. However, it is well known that the skin cell walls of different grape varieties have different composition and structure, which may generate differences in the easiness of the extraction of the compounds of interest, mainly located inside the skin cells. 

Rio Segade et al. [5] correlated skin hardness with lower anthocyanin extraction. Navarro et al. [6] stated that phenolic compounds’ concentration in the must of grapes could be related to thickness of the epidermis and cuticle: the greater the thickness, the lower the phenolic extraction observed. Ortega-Regules et al. [7] also stated that those varieties with thicker cell walls and higher content of pectins and cellulose could present a higher difficulty in extracting phenolic compounds; Medina-Plaza et al. [8] reported that cell wall material (CWM) composition indicated that demethylation of pectin and low lignin content favored the release of phenolics.

The use of maceration enzymes (enzymatic cocktails including pectolytic, hemicellulitic and cellulitic activities), pulsed electric fields (PEF) or US, among other techniques, look forward to improving the extraction of compounds of interest by helping to degrade the skin cell walls, as it has been demonstrated for enzymes and PEF [9]. Different studies have shown that this effect on cell walls may depend on variety. In this way, varietal effects have been observed when maceration enzymes were used [5,10], probably due to the different cell wall structure [11]. In this regard, differences in the outcome of PEF were also observed in different varieties [12].

Thus, the present work focuses on the study of the composition and phenolic characterization of wines made from three varieties of red grapes treated with US (Monastrell, Syrah and Cabernet Sauvignon), trying to determine if differences in the outcome of the application of US, if any, could be related with the biochemical composition of the grape cell walls. 

## 2. Materials and Methods

### 2.1. Wine Samples

Grape from three red grape varieties (Monastrell M, Syrah S and Cabernet Sauvignon CS) was harvested in September 2021 at the optimum moment of ripeness (25° Brix). The grapes were immediately transported to the winery where the clusters were destemmed and crushed. Two types of vinification were carried out for each variety: a control without US treatment (C) and another vinification carried out with sonicated grapes (US). For the application of US, an industrial-scale sonication device (Ultrawine, Agrovin S.A., Alcázar de San Juan, Spain) equipped with two hexagonal sonoreactors with several adhered sonoplates was used. The equipment worked at a frequency of 30 kHz, a power of 9000 W and a power density of 58.5 Wcm^-2^. The design of the US equipment and low residence time of the must in the system allow for maintaining the must temperature. The crushed grapes, both non-treated and treated ones, were introduced into 50 L stainless steel tanks, the total acidity was corrected (5.5 g/L), and commercial yeast *Saccharomyces cerevisiae* (Viniferm CT007, Agrovin, Alcázar de San Juan, Spain) was inoculated at a dose of 30 g/hL. The fermentation was controlled at 24 ± 2 °C. The skin maceration lasted 7 days, then the grape solid parts were pressed at 2 bars using a 75 L capacity pneumatic press. After the end of the fermentation, the wines were racked, removed from the lees, sulfited and cold-stabilized. After a month, the wines were again racked, sulfited and bottled. Analyses were performed one month after bottling. All the experiments were performed in triplicate. 

### 2.2. Skin Characterization by Optical Microscopy

For the analysis of grape skin by optical microscopy, the skin was sampled after crushing (control grapes) or after crushing and sonication (in the case of treated samples). Small pieces of tissue (1 mm^2^) were fixed in McDowell reagent (25% *v*/*v* glutaraldehyde and 40% *v*/*v* formaldehyde in 0.2 M cacodylate buffer) for 24 h at 4 °C, after which they were washed with a solution of cacodylate buffer and sucrose to remove the fixative reagent. A post-fixation in osmium tetroxide solution was performed for 2.5 h at 4 °C, and veronal uranyl acetate was used to perform the staining of the samples, applied for 2 h at 4 °C. Subsequently, the samples were dehydrated by a series of washes with gradients of increasing ethanol concentration (from 30% to 100% *v*/*v*), after which they were embedded in SPURR-type resin. Semi-fine sections were made and stained with toluidine blue and observed by optical microscopy.

### 2.3. Analysis of Cell Wall Material

#### 2.3.1. Isolation of Cell Wall Material

Cell wall material (CWM) of skins or pomaces was isolated following the procedure proposed by De Vries et al. [13]. Briefly, skins or pomaces (15 g) were suspended in water at 100 °C for 5 min and then homogenized. The homogenized material was mixed with two parts of 96% ethanol and extracted for 30 min at 40 °C. Then, the solid material was separated by centrifugation and extracted several times with fresh 70% ethanol for 30 min at 40 °C to remove soluble compounds. The alcohol-insoluble solids were then washed once with 96% ethanol and twice with acetone and dried under an air stream at room temperature.

#### 2.3.2. Analysis of Cell Wall Composition

Uronic acids were measured in the sulfuric acid cell wall hydrosilate using the colorimetric 3,5-dimethylphenol test after pretreating the cell walls with aq. 72% sulfuric acid for 1 h at 30 °C, followed by hydrolysis with 1 M sulfuric acid for 3 h at 100 °C. As a standard, pure galacturonic acid was used. After extraction of the CWM with 1 M NaOH (100 °C, 10 min), the protein and total phenolic compound content were measured using the colorimetric Coomassie Brilliant Blue assay and the colorimetric Folin-Ciocalteau reagent assay, respectively. Both pure gallic acid and bovine serum albumin (BSA) fraction V were employed as standards.

After pretreatment (30 °C, 1 h) of CWM with aqueous 72% sulfuric acid following of hydrolysis (100 °C, 3 h) with 1 M sulfuric acid, the total glucose was measured using a kit for glucose enzymatic analysis from R-biopharm (Darmstadt, Germany). Non-cellulosic glucose was measured after hydrolysis at 100 °C for 3 h using only 1 M sulfuric acid. The difference between the contents of total and non-cellulosic glucose was used to determine the cellulosic glucose. The amount of lignin was estimated using the acid-insoluble residue obtained after pretreatment and hydrolysis (Klason lignin).

### 2.4. Wine Physico-Chemical Analysis

The alcohol content, pH, total acidity and volatile acidity were determined in accordance with ECC regulations [14].

### 2.5. Wine Spectrophotometric Parameters

From filtered wine samples (using 0.45 μm nylon filters), the different analyses of chromatic parameters were performed using a HEλIOS α spectrophotometer (ThermoSpectronic, Thermo Fisher Scientific, Madrid, Spain). The color intensity (CI) was calculated from the sum of the absorbances obtained at 420, 520 and 620 nm [15]. The total polyphenol index (TPI) was determined at 280 nm according to the method of Ribéreau-Gayon et al. [16]. The determination of total and polymeric anthocyanins (Tant and Polant) and that of methylcellulose precipitable tannins (MCPT) were carried out by the methods described by Ho et al. [17] and Smith [18], respectively. 

### 2.6. Determination of Tannins by HPLC

For the determination of the tannic concentration and composition of the wines by high-performance liquid chromatography (HPLC), the phloroglucinolysis method and chromatographic analysis were carried out following the methodology described by Busse-Valverde et al. [19]. The identification and quantification of the adducts and flavan-3-ols were carried out at 280 nm. The chromatographic data allowed us to determine the concentration of total tannins (TT), the mean degree of polymerization (mDP), and concentration of epigallocatechin (EGC) and epicatechin gallate (ECG) subunits.

### 2.7. Identification and Quantification of Monosaccharides by GC–MS

Trimethylsilyl-ester O-methyl glycosyl residues produced after acidic methanolysis and derivatization as previously described were subjected to GC-MS analysis to evaluate the monosaccharide composition [20]. TMS refers to the total monosaccharides of precipitated polysaccharides. Based on the concentration of certain glycosyl residues, which are indicative of structurally recognized must and wine polysaccharides [21,22], the content of each polysaccharide family in the wine samples was determined. The total amount of mannoproteins (MP), rhamnogalacturonans type II (RG-II) and polysaccharides rich in arabinose and galactose (PRAG) was used to estimate the amount of total soluble polysaccharides (TSP). For the identification and quantification of monosaccharides, the methodology proposed by Gualdalupe et al. [21] was followed. 

### 2.8. Statistical Analysis

To determine whether there were any differences between the different vinifications, one-way analysis of variance with post-hoc Duncan (*p* < 0.05) was carried out. This analysis, together with a principal component analysis, was performed using the statistical program Statgraphics Centurion XVI (Statpoint Technologies Inc., Warrenton, VA, USA).

## 3. Results and Discussion

### 3.1. Wine Chemical Composition

The results of the physicochemical parameters of the wines obtained from control or sonicated grapes from the three different varieties are shown in Table 1. The effect of US on the wines’ chemical composition was very similar among varieties. Sonication generated a slight rise in pH and a slight decrease in total acidity (mostly due to a higher extraction of potassium from crushed grapes) and an increase in volatile acidity in Monastrell wines. Moreover, the Monastrell wines presented a higher alcoholic degree than the rest of the varieties, probably due to a greater release of fermentable sugars. Along with this, a higher content of total and residual sugars was observed in the sonicated Monastrell wine.

### 3.2. Chromatic and Phenolic Composition of the Different Wines

The results of the chromatic parameters of the wines obtained from control or sonicated grapes from the three different varieties are shown in Table 2.

Observing the percentage of increase caused by the application of US with the different chromatic parameters, and for the three different varieties, we can observe how the application of US resulted in an increase of the total phenol content of 20 and 26% for Syrah and Cabernet Sauvignon wines, respectively, and 6% for the Monastrell wines. Similarly, the increase in MCPT was 32% for Syrah and Cabernet Sauvignon wines and only 20% in Monastrell wines. 

We also studied the tannin profile of the three wines, both by the phloroglucinolysis methodology (Table 3). Among control wines, Monastrell wines presented the highest concentration of depolymerizable tannins, although the wines from Syrah and Cabernet Sauvignon grapes presented the higher increase in depolymerizable tannins due to sonication, whereas differences in the concentration of these tannins due to sonication were not significant in Monastrell wines; in fact, CS-US7d reached the highest value of these compounds, even higher than the concentration found in M-US7d.

The mDP decreased in all the wines made from sonicated grapes when compared with their control wines, this decrease being significant for Monastrell and Syrah wines; it is possible that sonication also favored a release of seed tannins as indicated by the increase in the concentration observed in (-)-epicatechin gallate (ECG), although the values of (-)-epigallocatechin (EGC) also increased significantly in Syrah and Cabernet Sauvignon wines, pointing to an important release of skin tannins from sonicated grapes. The galloylated units showed an increase of 50% in the wines of Cabernet Sauvignon and Syrah, and the concentration of EGC subunits increased between 30% in Syrah wines and 36% in Cabernet Sauvignon, which may favor a decrease of astringency in wines since this subunit has been negatively correlated with astringency perception [23,24]. On the other hand, the subunits arising from skin tannins in Monastrell wines made from sonicated grapes only increased by 7%, coincident with the lower extractability shown in phenolic compounds for Monastrell wines. 

Looking for a possible explanation of the difference in the outcome of the sonication of the grapes from the different varieties, we focused our attention on the skin cell wall structure and composition of the different varieties. It has been previously demonstrated that there is a correlation between the structure and composition of grape cell walls and the extractability of phenolic compounds; and that enological techniques that affect the grape cell wall structure may favor phenolic extraction. In this way, it has been reported that the addition of maceration enzymes during vinification caused cell wall depectination and that the enzymes unraveled the cell walls, enabling better extraction of phenolic compounds [25], the pectolytic enzymes’ activity influencing the kinetics of phenolic compounds extraction [26]. Cholet et al. [9] found that with the application of pulsed electric fields of high strength, long duration and high energy, the consequences for parietal structures of the skin were very significant, the pectic and phenolic skeletons being largely disorganized, which facilitated polyphenol extraction. Similarly, the collapse of the cavitation bubbles produced by high-intensity US energy led to an explosion of energy that, when occurring near a cell, disrupted it and increased phenolic extraction [27,28].

However, these authors did not study if there is a varietal effect of these treatments and if differences in cell wall structure could lead to different outcomes of the technologies. Only the studies of Apolinar-Valiente et al. [11] pointed to the fact that the grape variety and its skin cell wall composition and morphology might influence the efficiency of maceration enzymes on phenolic extractability, finding differences when comparing the enzymatic degradation of Syrah and Cabernet Sauvignon grapes. 

Our hypothesis is that the extension of the cell wall disruption caused by US may also vary depending on cell wall composition and structure, thicker cell walls being probably more difficulty degraded by US. Therefore, we studied the characteristics of the skin cell walls of the three varieties used in this work, both fresh grape skins and macerated skins, in order to determine if their cell wall characteristics could explain the observed differences in the outcome of US.

### 3.3. Optical Analysis and Composition Study of Grape Skin Cell Walls

Figure 1 shows the structure of the fresh skin of the three varieties observed by optical microscopy. The results of the analysis of these cell walls from the different varieties are shown in Table 4.

The optical microscopy allowed for evaluating the morphology of the cells of the most external layers of grape skins (Figure 1). The difference in the number of cell layers from cuticle to pulp in the different varieties can be clearly seen. The epidermis appeared as neat regular cells with moderately thick walls covered by a cuticle, followed by a second layer, the hypodermis, formed with a variable number of cell layers with thinner cell walls and with cell size increasing towards the pulp. Monastrell grapes showed a higher number of layers of skin cells, which presented thicker walls than those in the other varieties, even as the grapes were technologically mature, and the integrity of the skin cellular structure seemed to be maintained. Cabernet Sauvignon skin cells also presented quite thick cell walls, although, in general, integrity seemed to be more compromised. Syrah grapes presented, at the moment of harvest, thinner cell walls and a smaller transition from skin cells to pulp. 

These morphological differences may also be confirmed by looking at the quantity and composition of isolated cell wall material in fresh grapes (CWM, Table 4). Monastrell skins showed the highest quantity of CWM, while Cabernet Sauvignon and Syrah had a lower quantity. These results are totally coincident with those of Ortega-Regules et al. [29] in studying the same varieties.

The largest quantities of CWM found in Monastrell skins could indicate a stronger barrier for the extraction of the compounds located inside these cells, as previously stated by Ortega-Regules et al. [7] and Medina-Plaza et al. [30].

Cabernet Sauvignon cell walls are characterized by the lowest values of proteins and the highest values in pectins, measured as uronic acids. This variety, together with Monastrell, have a higher cellulosic glucose content in its walls, in accordance with the results obtained by Ortega-Regules et al. [7].

Lignin was significantly higher in Syrah skin cell walls and similar in those of Monastrell and Cabernet Sauvignon. Medina-Plaza et al. [30] observed that, of all the CWM components analyzed in Pinot Noir and Cabernet Sauvignon skin cell walls, only lignin and the amount of cell wall isolated were found to have a significant impact on phenolic extractability, and although our results are coincident when referring to cell wall amount, they were not regarding lignin content. Therefore, the composition and structure of fresh skin cell walls seem to be determinants of the effect of US and may explain the lower effect of US on the Monastrell chromatic characteristics. 

To gain more information on how US affects the degradation of cell walls and therefore the phenolic extractability, the skins from the control vinification and the vinification made with sonicated grapes were collected after a fermentative maceration of seven days and their cell walls analyzed (Table 4).

One of the most important changes observed comparing the data from fresh skins and those obtained after 7 days of fermentative maceration is the amount of isolated cell wall material, which was higher in macerated skins than in fresh grape skins, both for control and US macerated skins, coincident with the observations of Romero Cascales et al. [31] and Apolinar Valiente et al. [32]. This could be explained by the large degradation of the cell walls during the winemaking process. As stated by Apolinar Valiente et al. [32], a substantial loss in fresh mass caused by both the extraction of large amounts of soluble material and some dehydration has been proposed as the cause of this behavior during the vinification process. The isolated CWM was even higher in those skins that had been previously sonicated. One possible explanation could be that, when US is used, the inner content of the cells is even more easily extracted, an effect that can be confirmed with the chromatic improvement observed in the wines from sonicated grapes. 

Prithani and Dash [33] observed that ultrasound induced the formation of microscopic channels in the fruit structure and the US treatment increased water diffusivity due to the formation of these microscopic channels. Similarly, Pieczywek et al. [34] stated that ultrasound can affect any element of the tissue sub-structure and it can lead to cell-to-cell detachment, which increases the space between cells and loosens cell wall assembly. Rodrigues et al. [35] also stated that ultrasound induced the loss of cellular adhesion, the formation of large cell interspaces, light rupture of the cell walls and the formation of channels caused by rupture of the cell walls. The effect of US on the skin structure after 7 days of maceration can be observed in Figure 2.

It can be clearly seen how the skin structure is more disassembled when US was used; the reported cell-to-cell detachment, an increase in the space between cells and a loosening of cell wall assembly can be observed. 

Therefore, from the results of Prithani and Dash [33] and Pieczywek et al. [34], it seems that the increase in the diffusion of the intracellular content of skin cells to the surrounded media is more related to the formation of channels than to an actual degradation of cell walls. This was confirmed when the composition of the macerated skin cell walls was studied, since it barely changed from fresh tissue to macerated skins and from macerated skins from the vinification of controls to those from sonicated grapes. This behavior is different than that observed with the use of maceration enzymes. Apolinar-Valiente et al. [32] found that the uronic acids and total sugar content of the cell wall were lower for all the macerated skins from wines where a macerating enzyme was used due to the action of the enzymes on the pectin matrix.

### 3.4. Monosaccharide and Polysaccharide Composition of the Produced Wines

Other compounds of large enological importance and whose presence may be also related to skin cell wall composition and to the application of US are the soluble wine polysaccharides. Their origin is the skin and pulp cell walls. Vidal et al. [36] determined that 75% of the grape berry walls originate from the skin tissue; therefore, the composition of skin cell walls of the different varieties and the sonication of the grapes will influence the amount and composition of soluble polysaccharides in their corresponding wines. 

Figure 3 shows the total monosaccharides and polysaccharides families in the different wines. Related to cell wall composition, varietal differences were observed, Monastrell wines showed a significantly higher content of total monosaccharides components of the polysaccharides (TMS); rhamnogalacturonan type II (RG-II); mannoproteins (MP); polysaccharides rich in arabinose and galactose (PRAG); and therefore, total soluble polysaccharides families (TSP) compared to Syrah and Cabernet Sauvignon wines. Syrah and Cabernet Sauvignon wines did not show significant differences in these polysaccharides families.

The higher concentration for all the studied polysaccharide families observed in Monastrell wines is probably related to the higher amounts of cell wall material measured in this variety. Gil Cortiella et al. [37] also concluded that Carménère grape skin thickness was an influential factor in the final composition of the wine, resulting in more concentrated wines in terms of polysaccharides.

PRAG were the most abundant polysaccharides in the three varietal wines (approximately 50% of TSP), followed by RG-II (approximately 29% of TSP) and MP (approximately 21% of TSP).

Ultrasound treatment of crushed grapes significantly increased the content of total monosaccharides and that of each pectic polysaccharides family in the resulting wines, except for manoproteins, when compared with their control wines. These results showed that regardless of the grape variety used, the sonomechanical effect of ultrasound promoted the liberation of the soluble fraction of cell wall polysaccharides. Similar results were previously reported by Martínez-Lapuente et al. [3] in wines from a Monastrell grape variety.

Mannoproteins (MP) are the most common yeast-derived polysaccharides in wine and are introduced into the wine matrix during fermentation and aging [38]. Ultrasound treatment of crushed–stemmed grapes did not modify the release of MP from yeast during fermentation–maceration. The higher MP content in the Monastrell wines was probably a consequence of the higher content of the skin cell material causing a higher degree of turbidity in the must-wine during alcoholic fermentation. Previous studies have shown that the release of MP from yeast into the wine matrix depends on both the yeast strain and the turbidity of the must [39].

Wines from sonicated grapes also showed a significantly higher content of total monosaccharide components of the polysaccharides (TMS, calculated by adding the concentrations of the constituent monosaccharides of the pectic and non-pectic polysaccharides) than their respective controls.

What is interesting to point out is that Monastrell wines from sonicated grapes, contrary to the results found for phenolic content, showed a more pronounced increased in their concentration of polysaccharides when grapes were sonicated. Monastrell control wines had PRAG + RG-II content 1.6 times higher than the Syrah and Cabernet Sauvignon control wines, and the higher amount of Monastrell cell walls boosted the effect of sonication as regards the extractability of PRAG + RG-II. In this way, the sum of PRAG and RG-II content of Monastrell wines from sonicated grapes was 1.9 times higher than that of Syrah wines from sonicated grapes and 2.4 times higher than that of Cabernet Sauvignon sonicated wines. These results demonstrated that the presence of large amounts of skin cell wall material (such as in the case of Monastrell) favored the extraction of soluble polysaccharides in the wine, and that the effect is much more pronounced if maceration is performed with sonicated grapes. Brandão et al. [40] showed that PRAG and RG-II were able to reduce the interactions between salivary proteins and tannins. Therefore, the sonication, by increasing the PRAG and RG-II content in the wines, could be an enological technique to modulate the sensation of astringency of wines from grape varieties with thicker skins.

A multivariate analysis such as principal component analysis could be useful to reduce all this information provided by the chromatic, phenolic and polysaccharide analysis variables and more clearly give us a confirmation of the differences in sonication outcome depending on grape variety. We obtained two principal components (PC) that explained 85% of the variability of the data (Figure 4).

This analysis allows us to graphically locate the wine samples in the plane defined by PC1 and PC2 and view how separated they are according to variety and sonication effect. Control and sonicated samples were mainly separated along PC2 and PC1 separated samples according to variety. According to the different weights of the variables, sonicated samples were located in the upper part of PC2 where mostly all the chromatic and phenolic variables, except mDP, presented the highest loadings, demonstrating the improvement on the chromatic characteristics of these wines due to sonication. Wines made from sonicated grapes from Syrah and Cabernet Sauvignon were far apart from their control wines, and this separation was much smaller for Monastrell wines, corroborating our previous observations. Monastrell wines were also clearly separated from Syrah and Cabernet Sauvignon wines along PC1, mostly due to their higher concentration in the different polysaccharide families.

## 4. Conclusions

This study reinforces previous results indicating that the application of ultrasound in the production of red wines is a very interesting strategy to increase their phenolic content, the results being confirmed for wines made with different varieties. However, our results also indicate that there are varietal differences in terms of the final effect caused by the application of ultrasound, and these differences are related to the structural and compositional differences observed in the cell walls of the grapes of the different varieties studied here, especially the amount of cell wall material. Regarding chromatic characteristics and phenolic compounds, the effect of ultrasound is enhanced in those varieties with thinner skin cell walls and less cell wall material, and the opposite is observed for the wine polysaccharide content, their content in varieties such as Monastrell being clearly favored due to grape sonication; this may positively affect wine bitterness and astringency.

## Figures and Tables

**Figure 1 foods-12-02236-f001:**
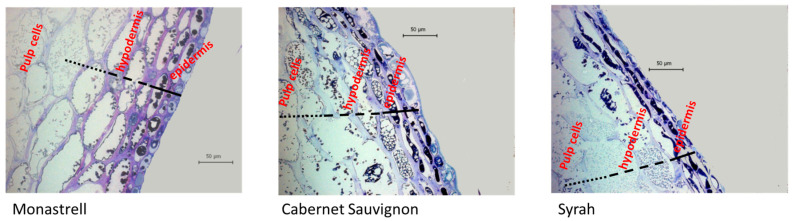
Optical microscopy of the skin of the three studied varieties.

**Figure 2 foods-12-02236-f002:**
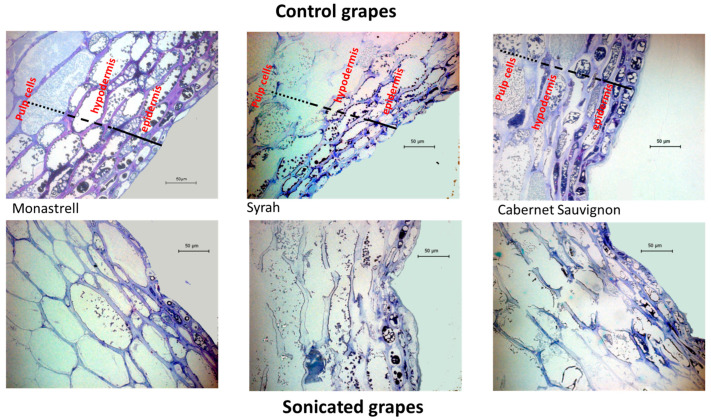
Optical microscopy of the control and sonicated grape skins after seven days of fermentative maceration.

**Figure 3 foods-12-02236-f003:**
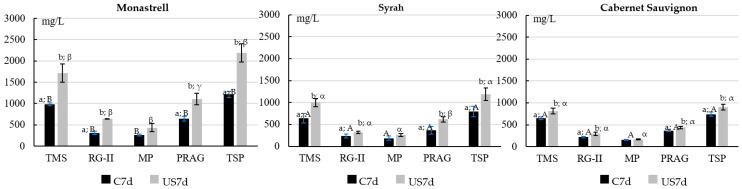
Total monosaccharides and polysaccharides families (mg/L) ^a^ in Monastrell, Syrah and Cabernet Sauvignon wines according to treatment. ^a^ Average of the three measurements. Different letters indicate statistical differences (*p* < 0.05). Lower-case letters compare treatment in the same monovarietal wine. Upper-case letters compare control wines. Greek alphabet letters compare wines with sonication treatment. TMS: total monosaccharides components of the precipitated polysaccharides; RG-II, rhamnogalacturonan type II; MP: mannoproteins; PRAG: polysaccharides rich in arabinose and galactose; TSP: total soluble polysaccharides families.

**Figure 4 foods-12-02236-f004:**
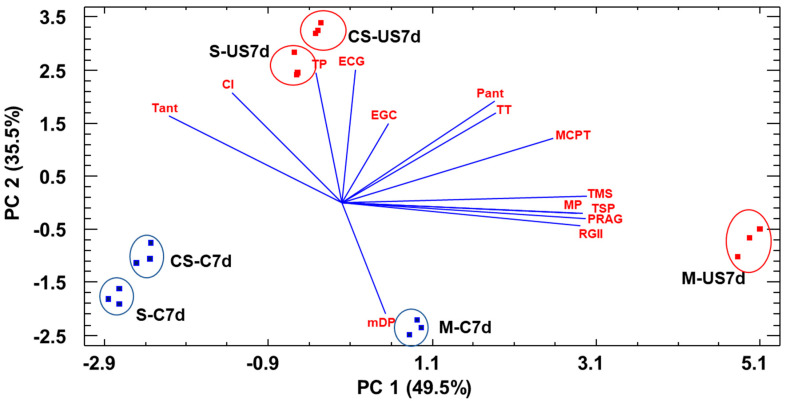
Distribution of the wine samples and weight of the different variables used in the principal component analysis. CI: color intensity; TP: total phenol content; Tant: total anthocyanins; Pant: polymeric anthocyanins; MCPT: total tannins (determined by the methylcellulose method); TT: total tannins determined by phloroglucinolysis; EGC: concentration of epigallocatechin subunits expressed as µM; ECG: concentration of epicatechin gallate subunits expressed as µM; mDP: mean degree of polymerization, TMS: total monosaccharides components of the precipitated polysaccharides; RG-II, rhamnogalacturonan type II; MP: mannoproteins; PRAG: polysaccharides rich in arabinose and galactose; TSP: total soluble polysaccharides families. M: Monastrell wines, S: Syrah wines, CS: Cabernet Sauvignon wines.

**Table 1 foods-12-02236-t001:** Physicochemical characteristics of the different wines.

	pH	TAc	VAc	G+F	TS	Gluc	Lact	Mal	°Alc
Monastrell									
M-C7d	3.74 a	4.98 b	0.56 a	0.70 a	1.33 a	0.18 a	0.11 a	1.54 a	14.93 b
M-US7d	4.00 b	4.77 a	0.74 b	1.15 b	1.80 b	0.20 a	0.10 a	1.60 a	14.61 a
Syrah									
S-C7d	3.80 a	5.55 a	0.25 a	0.49 b	1.07 a	0.10 a	0.11 a	2.23 a	13.50 b
S-US7d	3.90 b	5.65 a	0.24 a	0.20 a	0.97 a	0.07 a	0.13 a	2.76 b	13.10 a
Cabernet Sauvignon									
CS-C7d	3.79 a	5.69 a	0.32 a	0.37 a	0.87 a	0.04 a	0.14 a	1.97 a	13.24 a
CS-US7d	3.86 b	5.50 a	0.34 a	0.40 a	1.00 a	0.06 a	0.16 a	2.09 a	13.40 a

TAc, total acidity (g/L); VAc, volatile acidity (g/L); G+F, glucose + fructose (g/L); TS, total sugar content (g/L); Gluc, gluconic acid (g/L); Lact, lactic acid (g/L); Mal, malic acid (g/L), °Alc, alcohol percentage *v*/*v*. Different letters in the same column and for each type of wine mean statistically significant differences (*p* < 0.05) (*n* = 3 biological replicates for each variety).

**Table 2 foods-12-02236-t002:** Chromatic characteristics of the different wines.

	CI	TP	Tant	Pant	MCPT
	Monastrell				
M-C7d	10.9 a	50.1 a	406.2 a	47.5 a	1463.6 a
M-US7d	13.4 b	53.2 b	429.0 a	61.3 b	1850.4 b
	Syrah				
S-C7d	17.7 a	56.6 a	847.1 a	42.5 a	1051.6 a
S-US7d	21.6 b	71.2 b	883.2 a	60.6 b	1552.9 b
	Cabernet Sauvignon				
CS-C7d	15.7 a	50.2 a	700.5 a	48.9 a	1137.8 a
CS-US7d	18.9 b	67.9 b	845.2 b	59.2 b	1676.7 b

CI, color intensity; Tant, total anthocyanins (mg/L); TP, total polyphenol index; Pant, polymeric anthocyanins (mg/L); MCPT, total tannins per methylcellulose (mg/L). Different letters in the same column and for each type of wine mean statistically significant differences (*p* < 0.05) (*n* = 3 biological replicates for each variety).

**Table 3 foods-12-02236-t003:** Tannin concentration and composition in the three studied wines.

	TT	mDP	EGC	ECG
	Monastrell			
M-C7d	705.71 a	6.58 b	438.54 a	68.83 a
M-US7d	802.70 a	5.89 a	466.10 a	78.19 a
	Syrah			
S-C7d	418.10 a	6.09 b	286.57 a	62.09 a
S-US7d	757.65 b	4.31 a	401.60 b	124.95 b
	Cabernet Sauvignon			
CS-C7d	482.11 a	5.42 a	468.68 a	70.29 a
CS-US7d	922.35 b	5.33 a	731.40 b	134.43 b

TT, total depolymerizable tannins (mg/L); mDP: mean degree of polymerization; EGC: concentration of epigallocatechin subunits; ECG: concentration of epicatechin gallate subunits expressed as µM. Different letters in the same column and for each type of wine mean statistically significant differences (*p* < 0.05) (*n* = 3 biological replicates for each variety).

**Table 4 foods-12-02236-t004:** Amount of isolated cell wall material (mg/100 g of skin) and composition of major components in the cell wall from the three studied varieties (mg/g of cell wall), both in fresh grape skin and after seven days of fermentative maceration, for both control wine (C7d) and wine from sonicated grapes (US7d).

	CWM	Proteins	TP	UA	CG	NCG	Lignin
Monastrell							
Fresh grape	7.33	94.85 ± 1.74 a	80.51 ± 0.72 a	83.44 ± 1.98 a	157.08 ± 1.40 c	34.92 ± 1.49 a	549.19 ± 0.74 a
C7d	21.34	90.99 ± 0.17 a	82.65 ± 0.05 ab	94.06 ± 5.42 a	132.70 ± 1.01 a	48.67 ± 1.77 b	550.92 ± 5.08 a
US7d	23.67	94.36 ± 2.14 a	83.75 ± 1.06 b	90.98 ± 0.51 a	146.39 ± 1.96 b	47.09 ± 0.58 b	537.44 ± 2.81 a
Syrah							
Fresh grape	6.45	90.47 ± 2.82 b	73.68 ± 0.61 a	64.92 ± 1.03 a	129.50 ± 7.11 a	13.58 ± 2.21 a	627.84 ± 9.97 a
C7d	19.11	82.85 ± 0.91 a	86.20 ± 0.14 c	63.45 ± 3.78 a	121.72 ± 1.12 a	21.64 ± 0.95 b	624.14 ± 2.08 a
US7d	23.22	87.36 ± 0.93 ab	77.90 ± 0.71 b	71.54 ± 1.90 b	120.22 ± 4.42 a	19.06 ± 0.61 b	623.92 ± 7.39 a
Cabernet Sauvignon							
Fresh grape	6.48	79.13 ± 0.18 a	61.50 ± 1.93 a	102.93 ± 4.04 c	151.07 ± 0.78 c	12.91 ± 0.30 a	592.46 ± 4.59 a
C7d	17.50	88.59 ± 0.86 b	88.22 ± 1.75 b	63.45 ± 2.18 a	114.29 ± 1.41 a	15.70 ± 1.01 b	629.75 ± 2.89 c
US7d	22.25	88.18 ± 0.54 b	85.18 ± 0.51 b	73.75 ± 4.05 b	132.29 ± 1.22 b	13.31 ± 0.50 a	607.29 ± 5.49 b

Proteins are expressed as mg bovine serum albumin/g cell wall; total phenolic compounds (TP) as mg galic acid/g cell wall. UA: uronic acids; CG: cellulosic glucose; NCG: not cellulosic glucose. Different letters in the same column and for each type of cell wall material mean statistically significant differences (*p* < 0.05) (*n* = 3 biological replicates for each variety).

## Data Availability

Data is contained within the article.

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
