# Peer review of "High-Power Ultrasound in Enology: Is the Outcome of This Technique Dependent on Grape Variety?"

_foods, 2023, doi:10.3390/foods12112236_

Round 1

Reviewer 1 Report

In line 69: Were all the grapes 100% phytosanitary healthy and how was that determined?

In line 80: Please, write Saccharomyces cerevisae in italics. In line 83: What is the applied pressure and is it the same for all three varieties? In line 99: Reference? Is it a modified method? In line 150: “5mm).” is surplus. In line 254: highlight that kinetic of phenolic compounds extraction during fermentation could also depends of pectolytic enzymes activity. Kindly consider to cite Maced. J. Chem. Chem. Eng., 39, No. 2, (2020) 185–196.

In line 279: Mark the pictures with a.) Monastrell; b.) Syrah; c.) Cabernet Sauvignon.

Reviewer 2 Report

This work provides a detailed investigation on the effects of high power ultrasound and grape variety on the physicochemical composition of wine, which might be interesting for the improvement of wine quality. The experiments were well performed and the manuscript was organized logically. In my opinion, this manuscript need revisions.

1)      It is better to provide the physicochemical compositions of the grape berries for preparing wine samples.

2)      How long did the ultrasound treatment last?

3)      Figure 1: the grape varieties should be noted.

4)      Some discussion should be added. For example, the reason for rise in pH after ultrasound treatment, the significant increase in malic acid in Syrah variety, etc.

Author Response

See attached documents

Reviewer 3 Report

Manuscript foods-2417258 is concerned with the application of high-power ultrasound to the red grape types Monastrell, Syrah, and Cabernet Sauvignon. The investigation appears to have been carried out with considerable care. However, there are a number of questions that should be answered.

First and foremost, the manuscript is far too long. The authors should severely reconsider their writing style because the manuscript as it stands is somewhat dull and runs the risk of turning off readers too quickly. E.g., L150-161, L164-172, L372-375, can be deleted. They don’t add anything new to the narrative.

Second, the authors must find a better approach to present their findings. Tables are not the greatest approach to illustrate differences between qualities and variations. I would have preferred a simple box plot graph, or even better a PCA. This should be the writers' top priority in their revised work.

M&M: If the protocols in paragraphs 2.2, 2.3.1, and 2.3.2 have previously been published, they should be shortened and replaced with a relevant citation.

L33: Is it better to use HPU instead of US for high power ultrasound? Otherwise, what is the point of mentioning 'high power...'?

L51: CWM?? Please, write it in words.

L54: maceration enzymes? Hemicellulose? Xylanase? Please, be more specific.

L86: experiences? Do the authors mean experiments?

Table 1: check letters in the tables. Sometimes ‘a’ appears as a , sometimes as a. Please, be consistent.

L198: ‘….(n = 3) within the same variety….’ Please, specify.

L200: there were differences, see M-US7d for VA, G+F, TS and Mal.

L242: reference is needed after ‘astringency in wines’.

L252: reference is needed after ‘phenolic extraction’.

L282-288: This information should also be included in Figure 1. It is currently unclear to whom the authors are referencing. Please provide names and arrows in Figures 1a, 1b, and 1c so that readers can clearly identify the various skin areas.

L317-329: this should be one paragraph only, no three.

L413: ‘interesting’ instead of ‘interested’

already mentioned in my comments

Round 2

Reviewer 3 Report

The manuscript has significantly improved. Two more comments which I think the authors did not sufficiently addressed (my remarks were perhaps not as clear). I strongly advise the authors to take as much time as they need to make the necessary changes to minimise confusion in some of the tables and figures.

Tables 1, 2, 3 & 4: '....differences (P < 0.05) (n = 3 for each variety).....' The authors need to specify if n = 3 refers to biological or analytical replicates. E.g. (n = 3 biological replicates for each variety).

Figures 1 & 2: To assist readers in making a connection between the text and the image, the authors should include the terms "epidermis" and "hypodermis" with relative arrows in each of the photos.

Author Response

Dear Reviewer, thank you for your nice words. According to your suggestions, we have added in all the tables the following note (n=3 biological replicates for each variety) and in Figure 1 and 2 epidermis, hypodermis and the transition to pulp have been marked.

I hope that now you find our manuscript acceptable for publication.

Best regards

Dr. Encarna Gómez Plaza